# The Impact of Mental Health Leadership on Teamwork in Healthcare Organizations: A Serial Mediation Study

**Giulia Paganin** [1] , **Marco De Angelis** [2] , **Edoardo Pische** [3], **Francesco Saverio Violante** [1] , **Dina Guglielmi** [3] and **Luca Pietrantoni** [2,*]

1    Department of Medical and Surgical Science, Alma Mater Studiorum, University of Bologna, 40126 Bologna, Italy; giulia.paganin2@unibo.it (G.P.); francesco.violante@unibo.it (F.S.V.)
2    Department of Psychology, Alma Mater Studiorum, University of Bologna, 40126 Bologna, Italy; marco.deangelis6@unibo.it
3    Department of Educational Science, Alma Mater Studiorum, University of Bologna, 40126 Bologna, Italy; dina.guglielmi@unibo.it (D.G.)
*    Correspondence: luca.pietrantoni@unibo.it

**Abstract:** Background: There is compelling evidence to suggest that leadership behaviour and teamwork are critical success factors in healthcare organisations facing increasingly complex demands and limited resources. Effective teamwork is essential to deliver high-quality care, requiring integrating different professionals in the healthcare sector. Leaders play a significant role in facilitating teamwork by managing conflicts and promoting cooperation among team members. The COVID-19 pandemic has further highlighted the importance of leadership in supporting the mental health and well-being of team members. Methods: A cross-lagged research design was used to examine the relationship between mental health-specific (MHS) leadership and teamwork. Participants were 118 healthcare professionals (76.3% female; 44.9% aged between 45 and 54 years old). Results: A serial mediation model was confirmed, showing an indirect effect of mental health leadership on teamwork through interpersonal conflict and cooperation. Conclusions: Effective (MHS) leadership can positively impact the teamwork of healthcare professionals, particularly during times of crisis.

**Keywords:** leadership behavior; teamwork; healthcare organizations; COVID-19 pandemic; mental health promotion

## 1. Introduction

The psychology of sustainability and sustainable development represents a unique and innovative approach to addressing the critical and complex challenges facing society today, especially for its primary focus on fostering well-being in organisations at different levels, from employees to the organisation as an over-arching architecture [1]. This requires a comprehensive examination of both individual and environmental factors that shape the internal psychological processes in decision making. Therefore, to address the pressing challenges of sustainability and sustainable development, it is necessary to adopt a holistic and proactive approach to understand the underlying mechanisms that influence human behaviour [1,2] to examine new forms of leadership to promote sustainability in organisations [3]. Within this scenario, every year, healthcare organisations invest a lot in understanding, developing, and implementing more productive and sustainable work practices [4]. In recent years, the knowledge that appropriate leadership and teamwork are essential success factors in the increasingly complicated healthcare system has grown [5]. In this context, collaboration and teamwork are of paramount importance in the healthcare field, as the provision of high-quality care requires the coordinated efforts of a diverse team of specialised professionals. The success of these efforts relies on the ability of team members to work together seamlessly and to be jointly committed to the well-being of the patient. By leveraging the strengths of each team member, it is possible to achieve

exceptional clinical outcomes. Indeed, it is widely acknowledged that the provision of comprehensive healthcare is a complex and multifaceted endeavour that a single individual cannot achieve. The division of labour among healthcare professionals is essential to delivering high-quality care [6]. To this end, effective leadership plays a crucial role in managing conflicts and fostering a cohesive team dynamic that is necessary for optimal performance [7].

The COVID-19 pandemic disrupted work routines with immediate and significant consequences on workplace settings and employee well-being around the country [8,9]. Especially in the nowadays rapidly changing and uncertain environment, leadership must be adaptable and responsive to the evolving needs of healthcare teams. This includes providing support and guidance in previously not considered essential areas, such as focusing on workers' mental health and well-being. Leaders can help create a mutual respect and trust climate, mitigating conflicts within their workgroups and promoting collaboration among team members. Research has shown that discord among individuals can reduce team cohesion and overall performance [10].

Moreover, conflicts can create distractions that require time and effort to resolve, which can delay the completion of tasks and even put a team's goals at risk [11]. Once developed, interpersonal conflict might, in turn, possess deleterious effects on the workers, such as interference with the team's performance [12]. Given the significance of teamwork in healthcare organisations, comprehensive empirical research is required to understand better the behaviours that make up teamwork and the conditions that hinder or encourage it [13].

The relationship between health-promoting leadership and teamwork has received limited attention in the literature, and currently, the body of research on the effect of health-promoting leadership is still growing [14]. In particular, specific studies on the promotion of mental health by leaders are lacking, even if leadership abilities' effects on mental health (mental health-specific leadership—MHS leadership) have been studied and documented [15]. For this reason, we rely on the existing literature on transformational leadership. According to ref. [16], transformational leadership differs from health-promoting leadership but also shares some similarities, including the individual consideration dimension of attending to the needs of the employees by listening to their problems or assisting in developing their strengths. In this regard, the current research aims to explore the time-sustained indirect influence of leadership on teamwork, as mediated by factors such as interpersonal conflict in the workplace and team cooperation. The novel contribution of this research is its potential to identify a novel leadership approach and new areas of focus for interventions that can improve teamwork in the complex and dynamic healthcare environment.

### 1.1. Health-Oriented Leadership

Leadership is essential in healthcare organisations because it plays a critical role in shaping the culture, direction, and performance of the organisation and teams. Effective leadership can help to create a positive work environment, improve patient outcomes and increase staff morale and retention [17]. In healthcare, leaders are responsible for setting goals and visions for the organisation and ensuring that resources are used effectively to achieve these goals. They must also be able to make difficult decisions and provide guidance and support to their teams.

In healthcare settings, the importance of an effective leadership style has been repeatedly emphasised [18]. Healthcare professionals work in a complex and demanding environment, and leaders play a key role in creating a supportive and collaborative working environment that enables staff to provide the best possible care to patients. It has been especially important during the COVID-19 pandemic to choose the right strategy for managing one's team in order to handle challenges better and care for patients [9,18,19].

However, leadership definitions and studies are often conducted to explain performance but fail to account for particular leadership behaviours and attitudes relevant to

employee health and well-being [20]. Health has a significant value for individuals, organisations, and their employees. The quality and approach of leadership practices are key determinants of workplace health [21] and should be given due consideration in research and practice.

Based on these assumptions, we can examine the role of a specific leadership strategy named "health-oriented leadership". The health-oriented leadership concept (HoL) can be broadly characterised as a behavioural and organisational health-preventive strategy that aims to promote mental and physical health [22]. The approach involved leaders who prioritise their own physical and mental health (behaviour prevention) and, through their communication, leadership style, and role modelling, also address the health of their team members (organisational prevention) [23]. HoL is also characterised by staff-centred aspects [24,25], such as establishing mental health-promoting work environments (e.g., [26]) and engaging in direct attentive communication and interaction with staff members in a participatory process (e.g., proactively addressing stressed-out staff members to find solutions or help with prioritising work tasks).

Currently, the body of research on the effect of health-promoting leadership is still growing [14]. In particular, specific studies on the promotion of mental health by leaders are lacking, even if leadership abilities' effects on mental health (mental health-specific leadership—MHS leadership) have been studied and documented [15]. For this reason, we rely on the existing literature on transformational leadership. According to [16], transformational leadership differs from health-promoting leadership but also shares some similarities, including the individual consideration dimension of attending to the needs of the employees by listening to their problems or assisting in developing their strengths.

### 1.2. Leadership and Interpersonal Conflict

Recruitment issues, lack of resources, poor communication skills, and work overload may contribute to complex interpersonal dynamics inside healthcare companies [27]. For this reason, workplace interpersonal conflict and its effects on the healthcare system and the workforce have been examined [12]. Several studies pointed out that leadership styles, particularly transformative leadership, characterised mainly through behaviours that promote the team, could show beneficial effects by fostering a social environment in which team members resolve their disputes cooperatively (e.g., Zhang et al. [28]). Effective leadership can also contribute to conflict avoidance [29]. Transformational leaders are expected to create a culture of trust and respect among their team members to minimise conflicts [10].

Several studies described interpersonal conflict as a large variety of interpersonally problematic behaviours, such as rudeness, yelling, or other acts of workplace mistreatment [30,31]. Interpersonal conflict is also defined as a dynamic process that emerges among people with unpleasant emotional responses to perceived disagreement [32]. According to previous studies, interpersonal conflict is an inevitable aspect of cooperation and interactions among team members [28,33].

### 1.3. Leadership and Coordination

It has previously been observed that leadership can serve as a vehicle for coordination [34]. For example, research has underlined that feature of transformational leadership could enhance team coordination by promoting a shared understanding of goals and fostering communication and collaboration among team members. According to Liu and Li [35], transformational leadership improves coordination by assisting followers in making sense of the team's objectives. Moreover, in the healthcare scientific literature, team coordination is one of the prerequisites of effective teamwork (e.g., Manser [36]). Team coordination refers to the management of interdependencies among tasks through controlled action and information flow to accomplish a shared goal [34]. The research supports the value of team coordination capabilities in healthcare (e.g., Wright et al., [37]). A proper level of coordination is crucial for teams across all professions, and previous studies have demon-

strated that poor team coordination can have detrimental effects on outcomes [38]. For instance, the quality and safety of patient care tend to be impacted by a lack of coordination among practitioners at different levels [36]. Moreover, coordination is essential for effective teamwork [39].

### 1.4. Leadership and Teamwork

Teamwork is defined as interactions between team members who combine and integrate their resources to accomplish job requirements [40]. In the healthcare setting, teamwork is essential as it requires the contributions of numerous specialised professionals and/or teams from different departments. Therefore, each individual involved must be aligned in their commitment to the patient's well-being. Due to the division of duty among healthcare specialists, no single person can provide complete treatment or care [6]. Although coordination and job articulation are essential to care for units, team configuration and roles are frequently unclear, unstructured, or unstable. These characteristics of care teams suggest that teamwork may be complex. Nevertheless, there has been little research on hospital teams and the variables that affect team effectiveness [41].

A significant body of literature examined the positive influence of leadership style on teamwork [42]. Indeed, a lack of support from the leadership has been identified as one of the primary contributors to the failure of teamwork [43]. The results of these studies carried out in the pre-pandemic era apply well to the present time, as being a solid team and pulling together as a group is one of the keys to success during an organisational crisis [44]. In addition to promoting high-quality interprofessional cooperation, leadership is crucial in ensuring effective teamwork [45]. Teamwork problems in healthcare are often the result of a lack of coordination [46,47]. Moreover, teamwork effectiveness is likely to be impacted by psychological and cultural issues, such as interpersonal conflict, which can stimulate or impede communication within healthcare teams [41].

### 1.5. Study Hypothesis

As we previously said, the body of research on HoL is still growing, particularly referring to (MHS) leadership. Considering the above empirical findings, the gap in the literature with respect to HoL, and the relevance of this topic to the application context and historical period, the present research aims to provide a novel approach to leadership. Therefore, in light of these premises, we hypothesised a time-sustained indirect effect of (MHS) leadership on teamwork via interpersonal conflict and team cooperation within a sample of Italian healthcare professionals.

The hypotheses of this study are as follows:

**Hypothesis H1.** *(MHS) leadership is positively and significantly directly associated with teamwork (a), interpersonal conflict (b), and team coordination (c).*

**Hypothesis H2.** *(MHS) leadership is positively and significantly indirectly associated with teamwork, via interpersonal conflict and team coordination, in series.*

Moreover, we added gender and the impact of COVID-19 as covariates. The link between gender and teamwork has often been considered in the scientific literature, as it appears that the gender of team members may have an influence on the perceived quality of the teamwork itself [48]. At the same time, there have been several studies in the past two years evaluating the impact of COVID-19 on the teamwork of healthcare workers.

## 2. Materials and Methods

### 2.1. Measures

Mental Health-Specific (MHS) Leadership. In the present study, mental health-oriented leadership refers to all those leadership behaviours that consider the mental health of employees over the long term. (MHL) items were adapted from Gurt et al. [49]. The



responses were collected through a 5-point Likert scale ranging from 1 (i.e., completely disagree) to 5 (i.e., completely agree). Examples of items were as follows: "My supervisor discusses topics related to mental health and well-being with us", "My supervisor invites me to contribute my experience towards the implementation of mental health and well-being promotion projects", or "My supervisor is reflecting on how to increase mental health and well-being at our department". Cronbach's α was found to be 0.95, suggesting adequate internal consistency.

Interpersonal Conflict at Work. Interpersonal conflict is defined as a wide variety of interpersonally inappropriate actions, such as rudeness, screaming, or other acts of workplace mistreatment. It was measured using a nine-item scale based on Friedman et al. [50]. An example of items is "Backbiting is a frequent occurrence". As for the response scale, a 5-point Likert scale ranging from 1 (i.e., not at all) to 5 (i.e., a lot) was used. Cronbach's alpha was discovered to be 0.89, indicating a satisfactory level of internal consistency.

Team Coordination. Team Coordination is the controlled management of the interdependencies between subtasks to accomplish a common objective. It was measured using a three-item scale based on Salanova et al. [51] (e.g., "We coordinate with one another to complete the necessary tasks"). As for the response scale, a 7-point frequency scale ranging from 0 (i.e., never) to 6 (i.e., always) was used. Results determined that Cronbach's alpha was 0.82, indicating a sufficient level of internal consistency.

Teamwork. Teamwork is defined as interactions between team members who mix their resources to accomplish job requirements. It was measured using a three-item scale based on the work of Salanova et al. [52]. The items were worded as follows: "Your team... accepts its members offering innovative and creative ideas". As for the response scale, a 7-point frequency scale ranging from 0 (i.e., never) to 6 (i.e., always) was used. Analysis indicated that Cronbach's alpha was 0.82, showing sufficient internal consistency.

Perceived Impact of COVID-19 Pandemic on Work. The perceived impact of the COVID-19 pandemic on the workforce is defined as the extent to which workers believed their daily work, motivation, and relationship with their direct manager were affected by the pandemic outbreak. This was measured using a three-item scale adapted from a scale explicitly developed for this study by the partners involved in the European project after a thorough review of existing COVID-19-related scales. The items were worded as follows: "How has COVID-19 affected you and your work situation?—My work motivation is . . . " and "Support from the immediate manager is". As for the response scale, a 5-point Likert scale ranging from 1 (i.e., significantly made worse) to 5 (i.e., significantly improved) was used. Cronbach's alpha was discovered to be 0.75, indicating an acceptable level of internal consistency.

### 2.2. Participants

The sample was composed of 118 participants. A total of 33.9% of participants were aged between 45 and 54 years old. As for gender, 76.4% of the participants identified themselves as female and 23.6% as male. All of them were employees of an Italian Health Authority. Participants belonged to three different departments: the Department of Medicine (36.4%), the Emergency Department (39.0%), and the Scientific Hospitals and Care Institutions (IRCCS; 22.0%; missing data = 3%). A total of 25% of the participants held supervisory or managerial positions, while 72% were health professionals (e.g., nurses and physicians; missing data = 6.8%).

### 2.3. Procedure

This study is part of a European project funded by the H2020 program. The project, H-WORK [53], aims to design and validate protocols and multilevel interventions for promoting mental health in the workplace. The project received ethical approval from various academic institutions involved. This study was explicitly approved by the Bioethics Committee of the Alma Mater Studiorum—University of Bologna, following ethical requirements (Prot. n. 0185076) and in compliance with the Declaration of Helsinki [54]

Data collection took place between March 2021 and January 2022. Given the longitudinal nature of this study, we asked participants at T1 to create a personal code that they inserted on each survey they completed. The code did not contain any information that would allow the subject to be identified. The data collection was carried out in accordance with the Declaration of Helsinki's ethical norms. The researchers' contact information was shared with participants to obtain clarification if they had any questions or concerns. While implementing the first questionnaires, Italy was experiencing another peak in COVID-19 spread. Specifically, the Emilia-Romagna region was entering in the red zone on 15 March 2021. The administration of the second questionnaire took place after the so-called 'second wave', when it seemed that the spread of the virus had slowed down as a result of the containment measures taken.

### 2.4. Statistical Analysis

The statistical analysis for this study was performed using SPSS version 28 and the PROCESS macro v3.5 [55]. The data were first checked for normality, kurtosis, and skewness. Cohen's criteria [56] were then used to determine the magnitude of the "small" (0.10), "mid" (0.30), and "large" (0.50) correlation effects.

The proposed serial mediation model was tested using the PROCESS macro to investigate the role of interpersonal conflict at work (ICW) in mediating the effect of mental health leadership on teamwork and whether ICW and team coordination serially mediated the relationship between mental health leadership and teamwork. This method, which involves a bootstrapping procedure [57] to evaluate the indirect relationship between predictor and criterion variables through the mediator, was chosen as an alternative to the traditional Baron and Kenny mediation tests [58].

Model 6 was used to test the serial mediation of mental health leadership (T1) on teamwork (T2) through the serial mediation of ICW (T2) and team coordination (T2). The direct and indirect effects were calculated simultaneously, and the indirect effect estimate was computed using a bootstrapping approach with 95% confidence intervals. When zero is not included in the confidence interval ($p < 0.05$), it can be concluded that the indirect effect is significantly different from zero and that the effect of the independent variable (mental health leadership) on the dependent variable (teamwork) is mediated by the proposed mediating variables (ICW and team coordination). The analysis was conducted using 5000 bootstraps resamples.

### 3. Results

*Descriptive Analysis*

Table 1 shows means, standard deviations, and intercorrelations for key variables. The direction of all significant correlations between the variables was as anticipated. Additionally, all scales showed an internal consistency (Cronbach's alpha) above the threshold of 0.70 [59], as seen on the diagonal of the table.

**Table 1.** Mean, frequencies, standard deviation, correlation, and Cronbach's alpha of considered variables.

| | Mean/ Freq. | SD | 1 | 2 | 3 | 4 | 5 | 6 |
|---|---|---|---|---|---|---|---|---|
| 1. Gender | 79% (F) | - | - | | | | | |
| 2. COVID-19 Impact on Work | 2.56 | 0.70 | −0.19 * | (0.75) | | | | |
| 3. Mental Health-Specific Leadership T1 (Mhsl1) | 1.12 | 0.94 | −0.02 | 0.04 | (0.95) | | | |
| 4. Interpersonal Conflict at Work T2 (ICW2) | 2.87 | 0.72 | 0.04 | −0.18 | −0.22 * | (0.89) | | |
| 5. Team Coordination T2 (TC2) | 3.76 | 1.15 | 0.06 | 0.14 | 0.21 * | −0.55 ** | (0.82) | |
| 6. Teamwork T2 (Tw2) | 3.51 | 1.16 | 0.01 | 0.20 * | 0.30 ** | −0.48 ** | 0.64 ** | (0.79) |

Note: Cronbach's on the diagonal. ** $p < 0.01$; * $p < 0.05$. Gender: female = 0, male = 1.

Figure 1 shows the direct effect coefficient. (MHS) Leadership has a negative and significant direct effect on interpersonal conflict at work (T2), but there is not any significant

impact on team coordination (T2) and teamwork (T2). Interpersonal conflict at work (T2) does not significantly influence teamwork (T2). However, team coordination shows a significant effect on teamwork (T2). Concerning covariates, gender shows no influence on the dependent variable, whereas the impact of COVID-19 has a significant effect.

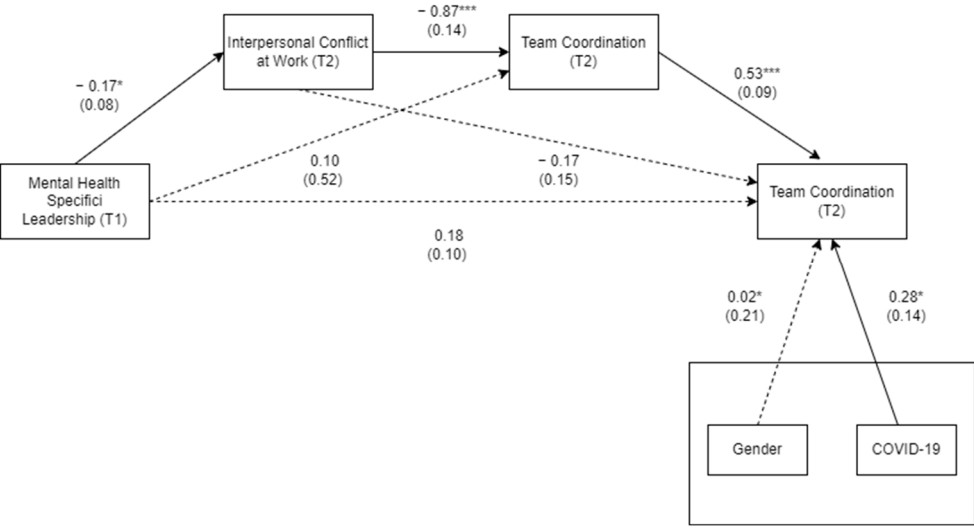

**Figure 1.** Hypothesised serial mediation model. Note: *** *p* < 0.001 and * *p* < 0.05. All parameter estimates are presented as standardised coefficients. Estimates (Est.). Standard error (SE). Confidence interval (CI).

Table 2 presents estimates of the indirect path coefficients and the 95% bias-corrected bootstrapped confidence intervals for our path estimations. Our findings demonstrate that interpersonal conflict at work (T2) and team coordination (T2), in series (estimate = 0.08; 95% CI = 0.01 to 0.18), wholly mediated the relationship between mental health-specific leadership (T1) and teamwork (T2), as zero was not included in the 95% confidence range.

**Table 2.** Indirect effect of hypothesised model.

| Indirect Effects Health-Promoting Leadership (HpL) | Est. | SE | CI 95% |
|---|---|---|---|
| MHSL → ICW → TW | 0.16 | 0.03 | (−0.02, 0.11) |
| MHSL → TC → TW | 0.03 | 0.08 | (−0.05, 0.17) |
| MHSL → ICW → TC → TW | 0.08 | 0.05 | (0.01, 0.18) |

Note: All parameter estimates are presented as standardised coefficients. Estimates (Est.). Standard error (SE). Confidence interval (CI). Mental health-specific leadership (MHSL). Interpersonal conflict at work (ICW). Team coordination (TC). Teamwork (TW).

## 4. Discussion

In recent years, psychological research has focused on studying complex psychological processes within individuals and environments from a sustainable perspective [1,2]. With the specific perspective of sustainable development of the personal and professional resources already present in the complex healthcare system, recent research has emphasised the importance of leadership in promoting effective teamwork among healthcare personnel to ensure positive patient outcomes [60]. The present study investigated the temporal, indirect effects of MHS leadership on teamwork through the sequential mediation of team cooperation and workplace interpersonal conflict. The potential contribution of this study is the identification of novel targets for future interventions in the complex healthcare context.

Our study results show that MHS leadership could directly decrease interpersonal conflict at work experienced by healthcare professionals, which in turn directly impaired team coordination. Finally, time coordination could improve teamwork between healthcare

professionals. Taken together, it can be posited that a leadership approach focused on mental health may have an indirect impact on teamwork by mitigating the potential adverse effects of interpersonal conflict in the workplace, thereby preserving the positive effects of team coordination on teamwork.

Concerning H1(a), our results did not confirm the direct effect of (MHS) leadership on teamwork, although several studies have confirmed the direct role of leadership. A direct effect of leadership was demonstrated, for instance, in a recent study that looked at the correlations among staff nurses' opinions of their nurse manager's leadership abilities, conflict management, and team backup on medical–surgical units [61].

Regarding H1(b), this study's results confirmed the negative direct effect of MHS leadership on interpersonal conflict. Our results align with the previous literature that underlines the impact of leadership on conflict. For example, a study by Kessler et al. [10] showed that transformational leadership was negatively and significantly associated with conflict with co-workers. Additional research on the relationship between managers' leadership practices and the types of conflicts that occur at work has also found that relational conflict is negatively impacted by inspirational leadership principles [62]. However, we did not find support for H1(c), which proposed a direct effect of leadership on team coordination.

The lack of a significant direct relationship between MHS leadership, team coordination, and teamwork may not be unexpected. This could be explained by the specific behaviours and elements of this leadership style, which is not primarily focused on clarifying job tasks or roles, but rather on fostering a team climate where individuals feel comfortable discussing or addressing any concerning aspects affecting their mental health. Therefore, this leadership style's direct and practical impact on coordination and teamwork may be observed as a distal and subsequent effect of the leader's efforts to promote mental health in the workplace. Further research investigating these underlying mechanisms is encouraged. Indeed, not many studies investigate leadership's direct effect on team coordination. However, other studies consider the link between leadership and coordination in team performance. For example, in a study by Hoch et al. [63], the authors investigate the effect of team coordination on the relationship between shared leadership and team performance. Effective coordination among team members facilitates the utilisation of their prior work skills. A leader must effectively manage these talents, especially in complex environments such as healthcare organisations. In their study, findings indicate that when shared leadership was lacking, coordination appeared to positively impact team results. Conversely, when coordination was inadequate, shared leadership was positively associated with team performance.

Regarding H2, we can confirm the hypothesis of the impact of MHS leadership on teamwork, serially mediated by interpersonal conflict at work and team coordination.

In general, leaders who adopt a health-promoting leadership approach can create a healthy working environment by regulating working conditions and decreasing health-related risk factors [64,65]. Health-specific leaders can help decrease conflicts and misunderstandings by ensuring that team members understand the goals and expectations of the team through open, attentive discussion and interaction with team members in a collaborative process [43,61,66]. It is well known that interpersonal conflict has been associated with worse staff and patient outcomes in healthcare teams [41]. Leaders proficient in managing and resolving conflicts can help reduce interpersonal conflicts within the team by providing support and guidance to team members. Additionally, leaders who can foster a sense of team cohesion and build strong relationships within the team can help reduce conflicts by creating a positive and supportive team environment [67]. The team leader can support staff members in creating shared goals and interpersonal relationships that reinforce cooperative goals [28].

There are various ways in which team coordination can increase teamwork. Firstly, when team members are well coordinated, they are more likely to communicate effectively, leading to improved teamwork [68]. Secondly, coordinated teams are often more efficient as members can work together smoothly and avoid duplicating efforts or interfering with

each other [69]. This can lead to faster completion of tasks and better overall teamwork. Lastly, when team members are able to work together effectively, they may develop greater trust in each other and be more likely to collaborate and support one another. This can foster a positive team dynamic and contribute to better teamwork.

Regarding the covariates, gender showed no direct effect on teamwork, whereas the impact of COVID-19 has a significant and direct effect. The latter effect has also been reported by other studies. For example, a study by Rehder et al. [70] reported that physicians noticed an increase in team climate during the first year of a pandemic. Arguably, in such a complex period, which forced healthcare workers to work even more closely together, it still had the merit of bringing the components of the various departments closer together, united in pursuit of a common goal.

This study has several limitations. This cross-lagged study was only carried out only in one hospital in Italy and only used self-report metrics. Furthermore, the data collection was carried out during a difficult period, although the impact of the variables was evaluated by controlling for the influence of COVID-19. However, it should be noted that the data collection periods were separated by a significant time span of almost one year, a long period of time during which various changes may have occurred. Therefore, future studies employing multiple methodologies and including nationally representative samples are encouraged, thus gathering data from several sources at different times to increase the robustness of the findings.

Another limitation is the common method bias [71], resulting from using the same evaluation method for both the dependent and the independent variable. In our case, the complexity of the context and the historical moment did not allow us to use different methodologies to measure perceived leadership and teamwork. However, future studies may benefit from different instruments [72] (e.g., teamwork perceptions observed by the leaders themselves). In addition, the scales used for this study have not yet been validated in Italian. Particularly with regard to the MHL scale, in the future, it might be very relevant to validate this scale for the Italian context.

Despite its limitations, this study had the merit of expanding the body of research regarding the importance of the health-oriented leadership approach. In this regard, health-promoting leadership seems to go further than just leadership styles, enhancing leaders' health consciousness and knowledge of the contextual drivers that impact it [14]. Beyond employing effective leadership techniques, leaders who support health have a specific focus on it and assume accountability for their and their team members' health [21,23]. In addition to boosting theoretical understanding, support for the hypotheses can have significant practical implications for improving leadership and teamwork, one of the most essential factors in the health sector. The results of this study emphasise the importance of considering variables at different levels, such as leadership and team levels. In order to maximise the impact of interventions, it is necessary not only to promote or change one organisational or personal aspect, but it is important to work on several sides [65]. For instance, this information can be used by healthcare organisations to design and deliver interventions aimed at educating leaders about the importance of health, including the mental health of their employees, showing its effect on various personal and organisational outcomes, such as reduced turnover and increased positive performance and health outcomes for patients. Currently, just a few leadership initiatives have undergone scientific evaluation and are intended to promote mental health in the healthcare industry. However, the interventions that target leadership appear to be the most promising approaches to addressing mental health in healthcare staff. Healthcare leaders who are aware of organisational and behavioural techniques are desperately needed to address the critical problem of mental health prevention in hospitals [45]. Of the same relevance is the need to train staff about effective conflict management and team cooperation strategies [61].

## 5. Conclusions

This article contributes to the growth of the body of research on health-promoting leadership, supporting the advancement of research from the perspective of the psychology of sustainability and sustainable development, focusing specifically on mental health promotion. In this period, characterised by the pandemic caused by the spread of COVID-19, a novel approach to leadership focused on promoting health, including mental health, is particularly useful for employees' well-being and increasing work performance. In a complex context, such as healthcare organisations undergoing enormous difficulties in this period, finding an effective leadership style fostering teamwork becomes particularly relevant. Several studies emphasise how leadership style can increase employee resources and decrease job risk factors. We have shown how leadership affects teamwork, mediated by interpersonal conflicts and team coordination. The leadership style focused on mental health contributes to decreasing the interpersonal conflicts experienced in the workplace, which usually decreases team coordination, an indispensable factor in the health sector. Coordination increases the perception of teamwork, an indispensable factor in several positive work outcomes. This study thus highlights how intervening in leadership in healthcare settings, shifting the focus not so much on performance but on the health of employees, including their mental health, can affect several factors that indirectly may impact the work outcomes of healthcare personnel that lead to more positive patient outcomes. Given the current research focus on health-oriented leadership on employees' health and well-being, future studies can explore the effects of this leadership style also on employee performance.

**Author Contributions:** Conceptualisation, G.P. and M.D.A.; methodology, G.P.; investigation, M.D.A.; resources, L.P.; data curation, G.P.; writing—original draft preparation, G.P.; writing—review and editing, L.P., E.P., F.S.V., D.G. and M.D.A.; supervision, L.P.; project administration, L.P. and M.D.A.; funding acquisition, L.P. All authors have read and agreed to the published version of the manuscript.

**Funding:** This paper has received funding from the European Union's Horizon 2020 research and innovation programme under the project H-WORK—Multilevel Interventions to Promote Mental Health in SMEs and Public Workplaces (grant agreement No 847386). The material presented and views expressed here are the responsibility of the authors only. The EU Commission takes no responsibility for any use made of the information set out.

**Institutional Review Board Statement:** This study received ethical approval from the Bioethics Committee of the Alma Mater Studiorum—University of Bologna (Prot. n. 0185076) and complied with the Declaration of Helsinki (World Medical Association, 2013).

**Informed Consent Statement:** Informed consent was obtained from all subjects involved in this study, outlining participation procedures, study content, data collection purposes, future data dissemination methods, participant rights, and contact information. Participation was voluntary, and participants had the option to withdraw at any time without repercussions. The data collected were anonymised, and only aggregated data were used in the analysis.

**Data Availability Statement:** The dataset is available from the corresponding author upon reasonable request.

**Conflicts of Interest:** The authors declare no conflict of interest.

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
