# Peer review of "The Impact of Mental Health Leadership on Teamwork in Healthcare Organizations: A Serial Mediation Study"

_sustainability, doi:10.3390/su15097337_

Round 1

Reviewer 1 Report

General comment

The authors presented a longitudinal study testing a model in which the relationships between Mental Health Specific Leadership and Teamwork was serially meditated by Interpersonal conflict at work and team Coordination. Furthermore, the authors insert in the model as covariates gender and COVID-19 Impact on Work. The study was run on 118 Italian workers employed in the healthcare sector. The Introduction covers sufficient ground to motivate the work. Methods are adequate and appropriate. Results are clearly displayed. The discussion starts from findings, and it is grounded in the literature. The result is a study of very good quality. Thus, there are only minor points for the authors.

Participants Section.

On page 5, line 232. The authors stated that “33.9% of participants were aged between 45 and 54 years old”. However, the mean age of participants is not declared. Please, indicate the mean age of participants, also adding the standard deviation.

Results.

On Page 6, line 284. The authors stated that “Additionally, all scales showed an internal consistency (Cronbach alpha's) above the threshold .65 [57].” Since the authors have cited DeVellis and Thorpe (2016), please see carefully at DeVellis and Thorpe (2016), on page 130: “Nunnally (1978) suggests a value of .70 as an acceptable lower bound for alpha. It is not unusual to see published scales with lower alphas. Different methodologists and investigators begin to squirm at different levels of alpha. Our personal comfort ranges for research scales are as follows: below .60, unacceptable; between .60 and .65, undesirable; between .65 and .70, minimally acceptable; between .70 and .80, respectable; between .80 and .90, very good; and much above .90, one should consider shortening the scale (see the following section).” I think that, following DeVellis and Thorpe (2016), the correct threshold is .70. After all, your data also fully satisfy this threshold.

Author Response

Authors' reply.

Reviewer 2 Report

Dear authors,

Thank you so much for summiting your enlightening research on such an exciting topic as new perspectives on understanding the leadership process. From my point of view, the manuscript reflects a very well-conducted study and provides new results regarding this leadership style that will undoubtedly receive more interest in the future.

Nevertheless, I would like to share some commentaries that I hope you will find helpful to improve even more the article:

1.       Introduction

·         The authors present the literature properly to understand the scope of the research. However, from my point of view, there is an aspect that affects the narrative of this section: in 1.1. Health-oriented Leadership is presented, but the field is still growing. For this reason, in the following paragraphs, transformative Leadership is taken as a basis for understanding the relationship between Leadership and other variables. Then, in the hypothesis subsection, HoL, or MHSL, appears again.

Due to the implicit aim of contributing to the insights about this leadership style and reinforcing that it is a relatively new perspective, it could be convenient to present it at the end of the Introduction section before the hypotheses. I reckon that it makes more explicit that the research is not a “leadership mediation model” but a novel approach to this process.

·         It should be positive a more detailed explanation of the hypotheses. Even in the previous paragraphs and the discussion, it is well developed; in 1.5, the expected direction of the relationships is unclear.

2.       Materials and Methods

·         From the redaction and the references section, I guess the scales used in this research are just an adaptation to the Italian language, but they are not validated, are they? If this is the case, it should be included as a limitation of the study, and maybe it could offer new research lines or future studies.

·         Gender and the perceived impact of the COVID-19 pandemic on work are included as covariates in the model. It should be helpful to have a brief justification.

·         In 2.2. some of the % do not sum 100%. Were they missing data, or were they typos?

·         In L237, do the authors mean “managerial”?

3.       Results

·         On L322-323 (Figure 1’ Note), some information seems not to fit (abbreviatures that do not appear in the figure)

·         L329: the upper limit of the CI is marked as .17, but in the table 2 is .18

4.       Discussion

·         Is the parenthesis required on “(MHS)” leadership? Please, check through the manuscript.

·         Some citations seem uncompleted when reading (e.g. L364: “a study by [10]; or L380: in a study by [61]). Please, check through the manuscript.

·         L478-480: please, check if all this information should be included in the final manuscript or if it is only a piece of advice for the authors.

Thank you so much,

Best regards

Author Response

Author's reply attached
